# Biochemistry, Pathophysiology, and Regulation of Linear Ubiquitination: Intricate Regulation by Coordinated Functions of the Associated Ligase and Deubiquitinase

**DOI:** 10.3390/cells10102706

**Published:** 2021-10-09

**Authors:** Yasuhiro Fuseya, Kazuhiro Iwai

**Affiliations:** Department of Molecular and Cellular Physiology, Graduate School of Medicine, Kyoto University, Sakyo-ku, Kyoto 606-8501, Japan; fuseya.yasuhiro@mcp.med.kyoto-u.ac.jp

**Keywords:** ubiquitin, linear ubiquitin chains, LUBAC, HOIL-1L, HOIP, OTULIN, NF-κB, cell death, selective autophagy, cancer

## Abstract

The ubiquitin system modulates protein functions by decorating target proteins with ubiquitin chains in most cases. Several types of ubiquitin chains exist, and chain type determines the mode of regulation of conjugated proteins. LUBAC is a ubiquitin ligase complex that specifically generates N-terminally Met1-linked linear ubiquitin chains. Although linear ubiquitin chains are much less abundant than other types of ubiquitin chains, they play pivotal roles in cell survival, proliferation, the immune response, and elimination of bacteria by selective autophagy. Because linear ubiquitin chains regulate inflammatory responses by controlling the proinflammatory transcription factor NF-κB and programmed cell death (including apoptosis and necroptosis), abnormal generation of linear chains can result in pathogenesis. LUBAC consists of HOIP, HOIL-1L, and SHARPIN; HOIP is the catalytic center for linear ubiquitination. LUBAC is unique in that it contains two different ubiquitin ligases, HOIP and HOIL-1L, in the same ligase complex. Furthermore, LUBAC constitutively interacts with the deubiquitinating enzymes (DUBs) OTULIN and CYLD, which cleave linear ubiquitin chains generated by LUBAC. In this review, we summarize the current status of linear ubiquitination research, and we discuss the intricate regulation of LUBAC-mediated linear ubiquitination by coordinate function of the HOIP and HOIL-1L ligases and OTULIN. Furthermore, we discuss therapeutic approaches to targeting LUBAC-mediated linear ubiquitin chains.

## 1. Introduction

Ubiquitin is a 76 amino acid (8.6 kDa) globular protein that is highly conserved in eukaryotic kingdoms. To exert its functions, ubiquitin must be conjugated to proteins via a cascade of reactions catalyzed by three types of enzymes: a ubiquitin-activating enzyme (E1), a ubiquitin-conjugating enzyme (ubiquitin carrier protein) (E2), and a ubiquitin ligase (E3) (Figure 1) [1]. The ubiquitin system was originally identified as part of an energy-dependent protein degradation system [1,2,3]. However, non-degradable roles of the ubiquitin system were first identified in 1995 [4], and we now know that the ubiquitin system is a sophisticated, reversible, post-translational protein modification system involved in the regulation of various physiological processes such as cell cycle, apoptosis, DNA repair, and signal transduction, in addition to protein degradation [5,6,7,8] (Figure 1). The most important feature of the ubiquitin system is that ubiquitin can be attached not only to its substrates but also to other ubiquitin molecules, thereby generating ubiquitin chains [5].

In most cases, the ubiquitin system modulates protein functions by decorating target proteins with ubiquitin chains. Several types of ubiquitin chains exist in cells. Previous work showed that ubiquitin chains are generated by adding ubiquitin to one of seven Lys (K) residues (K6, K11, K27, K29, K33, K48, and K63) on a ubiquitin molecule conjugated to a protein [5,9] (Figure 2). Because ubiquitin chains exert their functions by being recognized by binding proteins specific for each type of chain, the ubiquitin system can modulate protein functions in many ways. In 2006, Kirosako et al. firstly identified a novel linear ubiquitin chain that can be generated via the N-terminal Met (M) of ubiquitin [10,11,12,13].

In addition to ubiquitin chains composed of homologous linkages, heterotypic ubiquitin chains contain different linkage types; in addition, recent work showed that ubiquitin itself undergoes post-translational modification via phosphorylation, acetylation, sumoylation, and neddylation [14,15,16,17,18,19,20]. These findings further expanded the known roles of ubiquitin modifications. Cleavage of ubiquitin chains by deubiquitylating enzymes (DUBs) can terminate signals generated by ubiquitin conjugations. Thus, elucidating the mechanisms that underlie the ligation, recognition, and removal of ubiquitin chains is key to understanding ubiquitin chain functions [21].

In this review, we focus on N-terminally M1-linked linear ubiquitin chains, which are specifically generated by the linear ubiquitin chain assembly complex (LUBAC), the only E3 enzyme capable of generating such chains (Figure 3). Although linear ubiquitin chains are much less abundant than other types of ubiquitin chains, they play pivotal roles in cell survival, proliferation, the immune response, and elimination of bacteria by selective autophagy [11]. We will discuss therapeutic approaches that target LUBAC-mediated linear ubiquitin chains because abnormal generation of linear chains can result in pathogenesis [22]. In addition, we will discuss the intricate regulation of LUBAC-mediated linear ubiquitination via the coordinated function of ligases and DUBs [23], which provides new aspects in regulation of LUBAC functions.

## 2. Biochemistry of Linear Ubiquitin Chains

### 2.1. Linear Ubiquitin Chains Are Generated Specifically by the LUBAC Ligase Complex

The LUBAC E3 is composed of three subunits: HOIL-1L (large isoform of heme-oxidized iron regulatory protein2 (IRP2) ubiquitin ligase 1), HOIP (HOIL-1L interacting protein), and SHARPIN (SHANK-associated RH domain-interacting protein) [22,24,25,26] (Figure 3). LUBAC is unique because it contains two distinct RING-in-between-RING (RBR)-type ubiquitin ligase centers, one each in HOIP and HOIL-1L, within the same ubiquitin ligase complex. The RBR-type ubiquitin ligases recognize ubiquitin-bound E2 at their RING1 domain, transfer ubiquitin from E2 to a conserved cysteine (Cys) residue in the RING2 domain, and ultimately transfer it to substrate proteins or acceptor ubiquitin, thereby generating ubiquitin chains [27]. Of the two RBR centers in LUBAC, the RBR of HOIP is the catalytic center for linear ubiquitination. HOIP contains the linear ubiquitin chain-determining domain (LDD), located C-terminal to RING2, which is critical for linear ubiquitination. HOIP recognizes a ubiquitin moiety in the LDD domain that facilitates the transfer of ubiquitin from the conserved Cys in RING2 (Cys885 or Cys879 in human or mouse HOIP, respectively) to the α-amino group of the acceptor ubiquitin to form a linear linkage [28,29]. The RBR of HOIL-1L also has ubiquitin ligase activity; its roles in LUBAC will be discussed in Section 5.

### 2.2. Readers for Linear Ubiquitin Chains

To exert their functions, post-translational modifications must be recognized by binding proteins called “readers”. Because the type of ubiquitin chain determines the mode of protein regulation, ubiquitin linkages must be decoded by specific binding proteins in order to mediate their specific functions (Figure 4). To date, several domains have been identified as specific binders of linear ubiquitin chains: the UBAN domain in NF-κB essential modulator (NEMO) (also known as IKKγ); optineurin (OPTN) and A20-binding inhibitors of NF-κB (ABIN), including ABIN-1, ABIN-2, and ABIN-3; the NZF domain in HOIL-1L; and the seventh zinc finger (ZF7) domain in A20 (also called TNFAIP3) [30,31,32,33].

Among UBAN proteins, the most extensively studied is NEMO, a crucial regulator of the IκB kinase (IKK) complex [13]. Although NEMO UBAN was once thought to interact with K63-linked ubiquitin chains, it has a much higher affinity for linear ubiquitin chains. The NEMO UBAN binds to the hydrophobic patches centered at Ile44 and Phe4 of the distal and proximal parts of linear ubiquitin, respectively [30,34,35]. Recognition of linear chains by the UBAN domain of NEMO is critical for IKK activation, which induces canonical NF-κB activation [36].

The ABIN family consists of ABIN1 (also called TNIP1), ABIN2, and ABIN3, which are negative regulators of NF-κB signaling [37]. These proteins are involved in regulation of multiple signal transduction pathways including those involved in apoptosis, virus replication, and cancer progression. The crystal structures of the UBAN domains in ABIN1 and ABIN2 in complex linear ubiquitin chains have been solved [38,39]. Single-nucleotide polymorphisms (SNPs) of ABIN1 are associated with autoimmune disorders such as systemic lupus erythematous (SLE), Sjogren syndrome, systemic scleroderma, and psoriasis [40,41,42,43]. Furthermore, in mice, loss of the ubiquitin-binding activity of ABIN1 (ABIN1 D485N) or ABIN1 itself causes glomerulonephritis, which is characteristic of lupus nephritis with a high titer of pathogenic autoantibodies, including anti-nuclear and anti-double-stranded DNA antibodies. These observations imply that linear ubiquitination is involved in the pathogenesis of autoimmune disorders, especially SLE [44,45,46,47,48].

OPTN, a selective autophagy receptor, also interacts with linear chains specifically via its UBAN domain. OPTN recognizes linearly ubiquitinated cellular components such as impaired mitochondria or intracellular pathogens, including *Salmonella*, and eliminates them through OPTN-mediated autophagy. *OPTN* is also a causative gene for amyotrophic lateral sclerosis (ALS) and primary open-angle glaucoma (POAG) [49,50]. Moreover, ALS-associated OPTN mutants lose their ability to suppress NF-κB activation, mainly due to dysfunction of the UBAN domain in OPTN [51].

The HOIL-1L NZF domain, which specifically binds linear ubiquitin chains, is crucial for LUBAC-mediated canonical NF-κB activation [31]. A20 has a ZF7 domain, which specifically recognizes linear ubiquitin chains, and this part of the protein is indispensable for inhibition of LUBAC-mediated NF-κB activation [32].

### 2.3. Deubiquitinating Enzymes of Linear Ubiquitin Chains

Cleavage of ubiquitin chains conjugated to target proteins by deubiquitinating enzymes (DUBs) ceases the signaling elicited by ubiquitin chains [52,53] (Figure 1). In most reversible protein modification systems, such as phosphorylation, removing enzymes cut out modifiers from proteins, whereas some DUBs do not cleave whole ubiquitin modifications from proteins. More than 90 DUBs have been identified in humans, and some of these enzymes do indeed cleave whole ubiquitin modifications from proteins [52,53]. However, the ubiquitin system has a unique property: conjugation of ubiquitin chains regulates protein functions. Accordingly, DUBs that cleave specific inter-ubiquitin linkages, but not linkages between ubiquitin and substrate proteins, have been identified [52,53,54].

OTU deubiquitinase with linear linkage specificity (OTULIN) (also called FAM105B or Gumby) and cylindromatosis (CYLD) cleave linear ubiquitin chains, and both DUBs interact with LUBAC via the PUB domain of HOIP [55,56]. However, the binding systems differ in that OTULIN directly binds to HOIP via the PIM motif of OTULIN [55,57,58], whereas CYLD interacts with HOIP through spermatogenesis-associated 2 (SPATA2) [59,60,61,62]. Since both DUBs binds to the identical domain, the interaction should be mutually exclusive. However, further studies will be needed to elucidate precise binding modes of the two DUBs.

OTULIN is a DUB that specifically cleaves only inter-linear-ubiquitin linkages, but not substrate–ubiquitin bonds. In general, DUBs counteract ubiquitin ligases by cleaving ubiquitin chains. In other words, ubiquitin ligases turn the signal on, and then DUBs turn the signal off; however, OTULIN augments, but does not suppress, signals generated by linear ubiquitin chains [23,63]. The precise mechanism underlying this augmentation will be discussed in Section 5.

CYLD was identified as a DUB that specifically cleaves K63-linked chains, but it can also digest linear linkages. CYLD is the product of the causative gene in human cylindromatosis, a condition associated with multiple benign skin tumors [64], and is involved in the regulation of NF-κB activation [65,66]. Although the precise functions of CYLD in linear ubiquitination remain unknown, the absence of CYLD does not overtly increase the amount of linear ubiquitin chains; by contrast, the absence of OTULIN drastically increases the abundance of linear chains [67].

## 3. Structural Insights Regarding the LUBAC Ligase Complex

Recent advances in the structure of LUBAC are discussed in this section. Among three subunits of LUBAC, HOIP is the catalytic center for linear ubiquitination, whereas HOIL-1L and SHARPIN are also involved in the stabilization of the complex (Figure 3) [68]. In cells lacking HOIL-1L or SHARPIN, the amount of HOIP is drastically reduced because the complex is destabilized, leading to a significant decrease in the formation of linear ubiquitin chains. HOIP interacts with the ubiquitin-like (UBL) domains of the other two components. The UBL domains of HOIL-1L interact with the ubiquitin-associated (UBA) 2 domain of HOIP, and SHARPIN UBL interacts with HOIP UBA1 [68]. In addition to the interactions between HOIP and the other two subunits, recent biochemical and structural analyses revealed that the interaction between HOIL-1L and SHARPIN plays a pivotal role in stabilizing the trimeric LUBAC complex. Both HOIL-1L and SHARPIN have homologous LUBAC-tethering motifs (LTMs), consisting mainly of α-helices, N-terminal to their UBA domains. Surprisingly, the LTMs fold into a single globular domain [68]. Mutation or loss of the LTMs drastically destabilizes the complex, implying that LTM-mediated dimerization is critical for stabilizing LUBAC, possibly by folding into a single stable globular domain.

## 4. Physiological Functions of Linear Ubiquitin Chains

### 4.1. NF-κB Activation

LUBAC-mediated linear ubiquitination plays crucial roles in NF-κB activation and protection from programmed cell death [30,69,70] (Figure 5). First, we will discuss the molecular mechanism underlying NF-κB activation. NF-κB is a dimeric transcription factor consisting of five Rel homology domain (RHD)-containing proteins, including RelA (p65), RelB, c-Rel, p105/p50 (NF-κB1), and p100/p52 (NF-κB2). NF-κB is involved in a wide range of pivotal biological functions, including proliferation, the immune response, inflammation, and cell survival, and acts by binding to NF-κB-responsive elements referred to as κB sites [71]. Aberrant activation of NF-κB contributes to immunological disorders and oncogenesis [71,72,73]. Two pathways for NF-κB activation have been described, canonical and non-canonical; LUBAC participates in the former pathway [13].

The canonical NF-κB pathway is triggered by various stimuli such as TNF-α, IL-1β, CD40 ligand (CD40L), and ligands of Toll-like receptors (TLRs) [71]. LUBAC-mediated NF-κB activation has been extensively studied in TNF-α signaling [13] (Figure 5). Binding of TNF-α to TNF-receptor 1 (TNFR1) induces trimerization of TNFR and a conformational change in the intracellular death domain (DD) of TNFR1, which triggers recruitment of TNFR-associated death domain (TRADD) and receptor interacting serine/threonine-protein kinase 1 (RIPK1) to TNFR1 via direct interactions between the DDs. Next, TNF-receptor associated factor 2 (TRAF2) and cellular inhibitor of apoptosis proteins 1 and 2 (cIAP1/2) are recruited to TNFR1 to form TNFR-complex-I [11]. In the TNFR-complex-I, the cIAP ubiquitin ligases conjugate K63-linked ubiquitin chains to components of the TNFR-complex-I [11,12]. LUBAC is recruited to the TNFR-complex-I via recognition of K63 chains on the TNFR1 complex with the NZF domains of HOIP and SHARPIN [36,74]. LUBAC also recruits NEMO (the regulatory component of the IKK complex, which also contains IKK1 and IKK2) to TNFR-complex-I via recognition by the HOIP NZF1 domain and conjugates linear ubiquitin chains to NEMO [36]. Because the UBAN domain of NEMO interacts with linear chains with high affinity [34,75], the linear ubiquitin chains conjugated to NEMO are recognized by another NEMO, leading to activation of IKK2 via dimerization and trans-autophosphorylation of kinases in different IKK complexes, ultimately resulting in phosphorylation of inhibitor of κBα (IκBα) [71]. Phosphorylated IκBα is recognized by the SCF^βTrCP^ ubiquitin ligase, which conjugates K48-linked ubiquitin chains that target the protein for degradation by the proteasome [13]. The canonical NF-κB transcription factors then translocate into the nucleus and activate NF-κB target genes [71].

### 4.2. Cell Death Protection

LUBAC-mediated linear ubiquitination is also involved in protection from programmed cell death, including apoptosis and necroptosis induced by death receptors such as TNFR1 [76] (Figure 5). In TNF-α signaling, the death-inducible TNFR-complex-II, which consists of RIPK1, RIPK3, FAS-associated death domain protein (FADD), TRADD, and caspase-8 can be formed to trigger both apoptosis and necroptosis. LUBAC-mediated linear ubiquitination plays crucial roles in inhibition of TNFR-complex-II formation by conjugating linear chains to several components including RIPK1 and TNFR1 [77]. Although the precise mechanism underlying the conjugation of linear chains to proteins of TNFR-complex-I remains unclear, the process may involve a mixture of M1 and K63 chains, which are involved in the NF-κB activation pathway [78]. LUBAC can generate linear ubiquitin linkages by conjugating ubiquitin on the conserved Cys residue in RING2 to the α-amino group of ubiquitin recognized by the LDD domain [28,29]. Because the distal moiety of the K63 chain can be readily recognized by the LDD domain, K63 chains conjugated to proteins in TNFR-complex-I can act as a platform to conjugate linear chains to TNFR-complex-I to inhibit the formation of TNFR-complex-II.

## 5. Regulation of Linear Ubiquitination Activity of LUBAC

LUBAC, which generates linear ubiquitin chains, interacts with OTULIN, which eliminates them [55] (Figure 3). In addition to the HOIP E3 ligase center, which generates linear chains, LUBAC has another E3 center in HOIL-1L. Recent studies revealed the intricate regulation of LUBAC by the coordinated functions of HOIL-1L, HOIP, and OTULIN [23,63]. Conjugation of linear ubiquitin chains to all LUBAC subunits (auto-linear ubiquitination) inhibits the linear ubiquitination to other substrates (trans-linear ubiquitination) [63]. OTULIN maintains the linear ubiquitination activity of LUBAC by pruning the linear chains by binding to LUBAC via the HOIP PUB domain. The mechanism underlying auto-linear ubiquitination of LUBAC remains unknown, but functional analyses of HOIL-1L E3 have provided a clue. Fuseya et al. realized that mutations of the catalytic Cys residues of HOIL-1L RBR eliminated a slower-migrating HOIL-1L signal in immunoblots [23]. Further dissection revealed that HOIL-1L E3 mono-ubiquitinates all subunits of LUBAC. Because HOIP preferentially recognizes ubiquitin at the C-terminal LDD domain and conjugates ubiquitin to the α-amino group of the N-terminus of ubiquitin [28,29], ubiquitin conjugated to LUBAC subunits by HOIL-1L provides suitable substrates for the HOIP RBR, allowing conjugation of linear chains to LUBAC (auto-linear ubiquitination). Indeed, loss of E3 activity of HOIL-1L suppresses auto-linear ubiquitination of LUBAC almost completely and dramatically increases linear ubiquitination of target proteins (trans-linear ubiquitination) [23] (Figure 6).

Recently, Kelsall et al. showed that HOIL-1L can catalyze the formation of an oxy-ester bond between the C-terminal carboxyl group of ubiquitin and the hydroxyl groups of Serine (Ser) and/or Threonine (Thr) residues of substrate proteins [79,80]. However, HOIL-1L can mono-ubiquitinate a Lys residue in an artificial FLAG-tag added to N-terminus of HOIL-1L and that auto-linear ubiquitination of the Lys residue suppresses LUBAC functions, clearly indicating that auto-linear ubiquitination inhibits LUBAC function regardless of the position of the linearly ubiquitinated residues, including any Lys or Ser/Thr residues in LUBAC [23]. Some ubiquitin ligases, such as RNF213 and MycBP2 (also known as PHR1), are also able to catalyze the formation of an oxy-ester bond [81,82]. RNF213 directly conjugates ubiquitin to a non-proteinaceous substrate, the lipid A moiety of bacterial lipopolysaccharide (LPS), via formation of an oxy-ester bond [81]. Thus, oxy-ester ubiquitination may not be a unique feature of HOIL-1L, and the field awaits analyses of the physiological functions of oxy-ester ubiquitination.

Fuseya et al. clearly demonstrated the intricate regulation of the linear ubiquitination activity of LUBAC [23]. HOIL-1L E3 mono-ubiquitinates all LUBAC subunits, thereby facilitating HOIP-mediated conjugation of linear chains to LUBAC by providing a suitable substrate (i.e., ubiquitin) for HOIP E3, leading in turn to suppression of LUBAC functions. OTULIN counteracts these effects by cleaving linear chains from the LUBAC complex. Because LUBAC functions must be tightly regulated in cells, the main catalytic activity (HOIP E3) is regulated by the coordinated functions of the accessory E3 in the ligase complex (HOIL-1L) and DUB (Figure 6). It is very curious that auto-linear ubiquitination of LUBAC elicited by HOIL-1L E3 suppresses linear ubiquitination of target proteins. The molecular mechanism is currently unknown, but we speculate that auto-linear ubiquitination may cause HOIP RBR to preferentially recognize LUBAC itself as cis-targets over other substrates (trans-targets) for linear ubiquitination, which leads to suppression of LUBAC functions.

LUBAC is also regulated by cleavage of HOIL-1L [83,84,85]. HOIL-1L is cleaved by a paracaspase, mucosa-associated lymphoid tissue lymphoma translocation gene 1 (MALT1). MALT1, which removes HOIL-1L RBR domain from LUBAC by cleaving HOIL-1L between Arg165 and Gly166, is activated in most activated B-cell-like diffuse large B-cell lymphoma (ABC-DLBCL) [85]. The loss of E3 activity of HOIL-1L augments LUBAC functions, and that augmented LUBAC activity is associated with the pathogenesis of ABC-DLBCL [86,87]. Thus, MALT1-mediated cleavage of HOIL-1L might augment the functions of LUBAC, which plays critical roles in lymphomagenesis or resistance to chemotherapeutic agents.

## 6. LUBAC and Infections

### 6.1. LUBAC and Salmonella Infections

Recent work showed that LUBAC plays mandatory roles in elimination of pathogens such as *Salmonella* spp. [88,89,90,91]. *Salmonella* are Gram-negative, facultative, intracellular pathogens that invade host epithelial cells and macrophages [90] (Figure 7). In the early stage of infection, the invading bacteria reside in *Salmonella*-containing vacuoles (SCVs). After several hours, the SCVs rupture and *Salmonella* are exposed to the host cytosol, in which they can proliferate. However, cytosolic *Salmonella* are decorated by ubiquitin chains and targeted for autophagy (xenophagy) [90]. LUBAC is recruited to *Salmonella* by recognizing ubiquitin chains on the bacteria and then conjugates linear ubiquitin chains to the pre-existing ubiquitins. The resultant ubiquitin chains serve as a signaling platform. Linear ubiquitin chains on *Salmonella* recruit optineurin (OPTN) and induce xenophagy, ultimately leading to elimination of the bacteria [88,92].

Like NEMO, OPTN has a UBAN domain that selectively recognizes linear- and K63- ubiquitin chains [93]. Linear chains on *Salmonella* are also recognized by NEMO, which activates the IKK complex and NF-κB [88,89].

Loss of HOIL-1L E3 activity augments generation of linear ubiquitin chains by LUBAC and efficiently restricts proliferation of *Salmonella* as well as infection-induced cell death. Furthermore, in cells expressing a HOIL-1L mutant lacking E3 activity, high levels of linear ubiquitin chains conjugated onto *Salmonella* strongly activate NF-κB [23], implying that attenuation of HOIL-1L E3 activity is a promising therapeutic target for eliminating such bacterial pathogens by augmenting LUBAC functions.

As mentioned above, LUBAC is recruited to *Salmonella* by recognition of pre-existing ubiquitin coats on bacteria. Although the proteins that contribute to the initial step, the bacterial molecule modified by ubiquitin, and the enzyme that directly ubiquitinates *Salmonell*a have not been identified, recent work showed that RNF213 conjugates the first ubiquitin to bacteria [81] (Figure 7). Surprisingly, RNF213 directly conjugates ubiquitin to a non-proteinaceous substrate, the lipid A moiety of bacterial lipopolysaccharide (LPS). RNF213 is the largest known human E3 ligase (almost 600 kDa) and is the major susceptibility gene for moyamoya disease [94,95,96], a cerebrovascular disorder that is characterized by bilateral stenosis of the supraclinoid internal carotid artery and abnormal formation of collateral vessels. Ubiquitination of *Salmonella* by RNF213 leads to recruitment of LUBAC and restricts *Salmonella* proliferation by inducing xenophagy and NF-κB activation [81].

### 6.2. Suppression of Linear Ubiquitination by Pathogens

Some pathogens target LUBAC to facilitate their proliferation. Gliotoxin, a major virulence factor of the opportunistic pathogen *Aspergillus fumigatus*, is a specific inhibitor of LUBAC [97]. The fungal metabolite gliotoxin specifically inhibits LUBAC by binding to the RING-IBR-RING domain of HOIP, and inhibiting signal-induced NF-κB activation. This raises the possibility that LUBAC inhibitors could be isolated from natural products. Furthermore, some bacteria secrete effector proteins into host cells to facilitate their proliferation by modulating the functions of host proteins [91,98]. Many of these effectors target the ubiquitin systems, and some specifically target LUBAC. The entero-invasive bacterium *Shigella flexneri* secretes the effector protein IpaH1.4 into host cells [98]. IpaH 1.4, a ubiquitin ligase that directly interacts with the LUBAC subunits HOIL-1L and HOIP, catalyzes conjugation of K48-linked ubiquitin chains to the RING-IBR-RING domain of HOIP, leading to degradation of HOIP by the proteasome and a decrease in the level of LUBAC. As mentioned above (Section 6.1, LUBAC and *Salmonella* infections), LUBAC is recruited to the ubiquitin coats of cytosolic bacteria to generate linear ubiquitin chains on their surfaces, leading to restriction of bacterial growth though activation of autophagy and the NF-κB pathway [90] (Figure 7). IpaH1.4 secreted by *Shigella flexneri* inhibits the formation of linear ubiquitin chains on the surfaces of cytosolic bacteria by decreasing the level of LUBAC, enabling bacteria to escape from xenophagy.

Other pathogens secrete effector proteins that have deubiquitinase activity into host cells to disrupt the ubiquitin-mediated host defense system. The intracellular bacterium *Legionella pneumophila* secretes effectors that target linear ubiquitin chains [99]. *Legionella pneumophila* secrets RavD, which specifically cleaves linear ubiquitin chains. A RavD ortholog was identified in *L. clemsonensis*, and linear-ubiquitin-specific DUB activity was detected in lysates from *L. bozemanni*, suggesting that secretion of effectors with linear-ubiquitin-specific DUB activity is a general mechanism among *Legionella* species [91,99].

## 7. Linear Ubiquitination in Diseases

### 7.1. HOIP Deficiency in Mice and Human

Mutations of the ligase and the DUB for linear ubiquitination cause autoinflammatory diseases in humans. HOIP-knockout mice are embryonically lethal at approximately E10.5 and exhibit disrupted vasculature in the yolk sac [100].

In humans, two patients with HOIP deficiency have been identified in different families [101,102]. The first case of HOIP deficiency, an adolescent patient homozygous for the L72P missense mutation in the PUB domain of HOIP, presented with multiorgan autoinflammation, immunodeficiency, systemic lymphangiectasia, and subclinical amylopectinosis [101]. The second case, a child with the c.1197G > C and c.1737 + 3A > G mutations, has early-onset immunodeficiency and autoinflammation but not amylopectinosis and lymphangiectasia [102]. In both of these cases, the amount of HOIP was drastically reduced due to the mutations, and the symptoms were attributed to reduction in the levels of LUBAC.

### 7.2. HOIL-1L Deficiency in Mice and Humans

Mice lacking HOIL-1L exhibit embryonic lethality around E10.5, as in HOIP-knockout mice [68,103]. Human HOIL-1L deficiency is associated with immunodeficiency and autoinflammation; however, a substantial number of patients with mutations in HOIL-1L exhibit polyglucosan body myopathy/cardiomyopathy without immunological disorders [104,105]. The pathogenesis of polyglucosan accumulation has not been elucidated, but various mechanisms could be involved. In patients with HOIL-1L deficiency who lack immune symptoms, the mutations are located mainly in the C-terminal half of the protein, leading to the ligase activity of HOIL-1L (Figure 3). HOIL-1L interacts with HOIP and SHARPIN via the N-terminal region; consequently, patients with mutations in the C-terminal half of the protein have substantial amounts of LUBAC and linear ubiquitination activity, potentially explaining the lack of immunological symptoms.

### 7.3. SHARPIN Deficiency in Mice and Humans

To date, no patients with SHARPIN deficiency have been reported. Mice lacking SHARPIN exhibit chronic autoinflammation in the skin (chronic proliferative dermatitis in mice: cpdm) due to augmented TNF-α-induced death of keratinocytes, a result of the decrease in LUBAC ligase activity caused by reduced levels of HOIL-1L and HOIP [24,25,106,107]. In cpdm mice, introduction of even one HOIL-1L E3 ligase-dead allele dramatically ameliorates dermatitis and suppresses keratinocyte apoptosis without affecting the amount of HOIP [23]. This observation suggests that augmentation of linear ubiquitination activity of HOIP E3 by HOIL-1L lacking E3 ameliorates the symptoms of cpdm. Moreover, these findings indicate that cpdm is caused mainly by attenuation of HOIP E3 activity rather than altered subunit composition of LUBAC.

### 7.4. OTULIN Deficiency

OTULIN knock-in mice with a mutation in the active-site cysteine (C129A) exhibit embryonic lethality with abnormal vasculature at E10.5 (E10.5), as in HOIP and HOIL-1L knockout mice [63]. In humans, OTULIN deficiency results in development of OTULIN-related autoinflammatory syndrome (ORAS), which is associated with recurrent fevers, autoantibodies, diarrhea, panniculitis, and arthritis [108,109,110]. Because OTULIN prevents auto-linear ubiquitination of LUBAC and maintains the LUBAC activity [23,63], OTULIN deficiency induces deterioration of LUBAC.

### 7.5. Augmentation of LUBAC Activity in Cancer

LUBAC-mediated linear ubiquitination plays crucial roles in NF-κB activation and protection from cell death, both of which are associated with oncogenesis [11]. Augmentation of LUBAC activity is shown to be associated with carcinogenesis. Rare germline SNPs in HOIP are significantly enriched in activated B-cell-like diffuse large B-cell lymphoma (ABC-DLBCL) [86]. ABC-DLBCL is characterized by constitutive NF-κB activation mediated by the B-cell receptor (BCR) and Toll-like receptor (TLR) signaling pathways, and many oncogenic mutations within these pathways have been identified [111,112,113,114,115]. The SNPs enriched in ABC-DLBCL patients induce the substitution of amino acids that increase linear ubiquitin chain formation by LUBAC, which augments NF-κB activation [86]. Furthermore, clinical RNA sequencing (RNA-seq) gene expression data revealed that expression of HOIP is elevated in human ABC-DLBCL [87]. To probe the involvement of augmented LUBAC activity in lymphomagenesis, mice overexpressing HOIP were generated [87]. Although augmented LUBAC activity did not induce B-cell lymphomagenesis, introduction of HOIP facilitated generation of B-cell lymphomas induced by oncogenic mutation of MyD88 [87]. Protection from cell death as well as NF-κB activation underlies facilitation of lymphomagenesis. Moreover thiolutin, a natural compound that inhibits LUBAC, suppresses the growth of B-cell lymphomas in a mouse transplantation model [87]. As mentioned above, it has been proposed that augmentation of LUBAC activity is associated with resistance to cancer therapies. LUBAC plays a role in resistance to a widely used anti-cancer drug cisplatin [116,117]. In squamous lung cells, enhanced LUBAC-mediated NF-κB activation appears to be a determinant of cis-platinum resistance [118]. Thus, inhibition of LUBAC represents a promising therapeutic strategy for not only malignant lymphoma, but also a broad spectrum of malignant tumors mainly by augmenting NF-κB activation.

## 8. Therapeutic Approaches to Targeting LUBAC

### 8.1. Cancer Therapy via Attenuation of LUBAC

As mentioned above (Section 7.5), augmentation of LUBAC is associated with carcinogenesis [87]. Hence, decreasing the level of LUBAC represents a promising therapeutic strategy for treating cancer. Several agents that inhibit LUBAC have been found. Gliotoxin, a fungal metabolite, was the first small molecule shown to inhibit linear ubiquitination activity [97]. Thiolutin and aureotricin, products of streptomycetes, also inhibit ligase activity [87]. However, these natural products are not specific for LUBAC. HOIPIN-8 is a synthetic agent that inhibits LUBAC linear ubiquitination by interacting specifically with HOIP [119]. However, considering that loss of LUBAC activity causes embryonic lethality in mice, compounds that inhibit the catalytic activity of LUBAC might be highly toxic. Accordingly, other strategies to decrease LUBAC activity than inhibition of the catalytic activity have been proposed. Among the three interactions between the LUBAC subunits, the LTM-mediated dimerization of HOIL-1L and SHARPIN appears to play the predominant role in stabilizing the complex [68]. LUBAC ligase activity is not completely abolished by disruption of the interaction between the two accessory subunits, as LUBAC containing HOIL-1L and HOIP or SHARPIN and HOIP can exist. Therefore, agents that target the dimerization of HOIL-1L and SHARPIN might have fewer side effects than those that inhibit the catalytic activity of HOIP. The crucial role of LTM-mediated heterodimerization of the two accessory subunits in stable formation of trimeric LUBAC suggests a therapeutic strategy for the treatment of malignant tumors. In addition to the crucial roles of LUBAC in the oncogenesis of ABC-DLBCL and resistance to cis-platinum [116,117,118], LUBAC activity is also involved in the resistance to anti-programmed death-1 (PD-1) therapy in murine B16F10 melanoma cells [116,117,120,121]. Therefore, development of LUBAC inhibitors with fewer side effects has been awaited.

### 8.2. Treatment of Infectious Disease via Augmentation of LUBAC

As mentioned above (Section 6), LUBAC plays pivotal roles in eliminations of pathogens, such as *Salmonella*, via linear ubiquitin-dependent selective autophagy, and some pathogens secreted effector proteins in order to destabilize LUBAC [90,91]. Furthermore, LUBAC is also involved in clearance of several viruses, including norovirus [122]. Thus, LUBAC has recently attracted a great deal of attention as a therapeutic target for infections; however, it remains unclear how to activate LUBAC functions. A recent study by our group showed that HOIL-1L inhibits LUBAC functions by mono-ubiquitinating all subunits of LUBAC, and that inhibition of E3 activity of HOIL-1L dramatically increases LUBAC functions [23]. Thus, the HOIL-1L E3 activity is a promising therapeutic target for augmenting LUBAC functions. Furthermore, since mice expressing a HOIL-1L mutant lacking E3 activity are viable up to the age of 12 months without overt phenotypes, and augmented HOIP expression failed to induce lymphomagenesis [87], agents that target the E3 activity of HOIL-1L could have fewer side effects.

## 9. Conclusions

LUBAC, the only ligase that can generate linear ubiquitin chains, plays pivotal roles in NF-κB activation, protection against cell death, and elimination of bacteria by induction of xenophagy. Moreover, deficiency of LUBAC components is associated with several disorders in humans (Appendix A). Consequently, LUBAC and linear ubiquitin chains are attracting intense research attention. LUBAC is a unique E3 because it contains two different ubiquitin ligase centers in the same ligase complex. A recent work revealed that the E3 activity of HOIL-1L plays a crucial role in LUBAC regulation. HOIL-1L conjugates mono-ubiquitin onto all LUBAC subunits, followed by HOIP-mediated conjugation of linear chains onto mono-ubiquitin; these linear chains attenuate LUBAC functions. Introduction of E3-defective HOIL-1L mutants augmented linear ubiquitination, protecting cells against *Salmonella* infection and curing dermatitis caused by reduction in LUBAC levels due to loss of SHARPIN. Thus, inhibition of the E3 activity of HOIL-1L E3 represents a promising strategy for treating severe infections or immunodeficiency.

## Figures and Tables

**Figure 1 cells-10-02706-f001:**
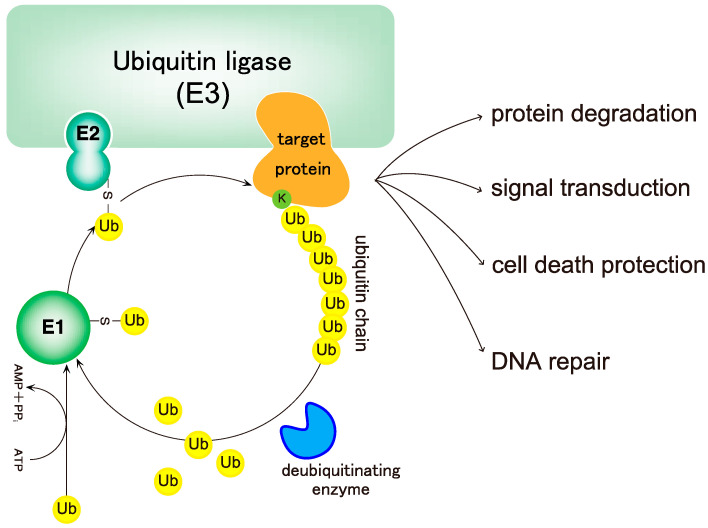
The ubiquitin system. Ubiquitin is conjugated to target proteins via E1 (ubiquitin-activating enzyme), E2 (ubiquitin-conjugating enzyme), and E3 (ubiquitin ligase) activities, leading to the conjugation of ubiquitin to substrates recognized by E3s. First, ubiquitin is conjugated onto a substrate, followed by the conjugation of ubiquitin moieties onto the distal end of the ubiquitin chain. Ubiquitination of proteins regulates various cellular functions depending on the type of ubiquitin linkage. Finally, ubiquitins are removed by deubiquitinases (DUBs), and ubiquitin monomers newly trimmed by DUBs are integrated into the ubiquitin pool to be used for ubiquitination of other proteins.

**Figure 2 cells-10-02706-f002:**
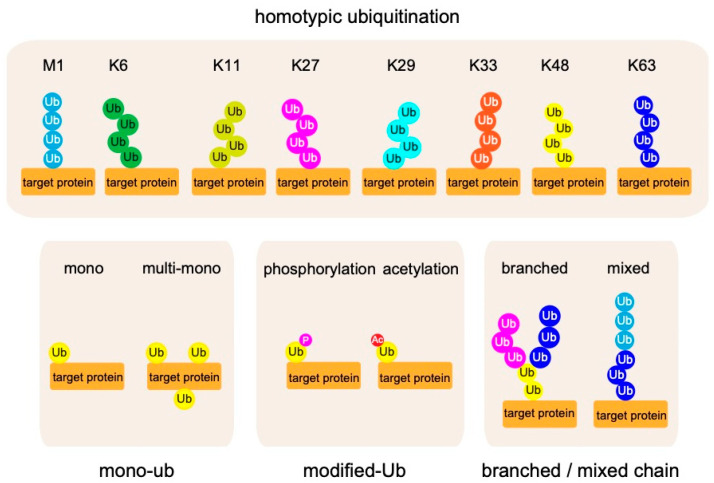
Complexity in the ubiquitin code. Eight types of homotypic ubiquitin linkages are known to exist: M1, K6, K11, K27, K29, K33, K48, and K63. Furthermore, mono-ubiquitination, post-translational modification of ubiquitin itself, and heterotypic ubiquitin chains containing different linkage types have recently been identified.

**Figure 3 cells-10-02706-f003:**
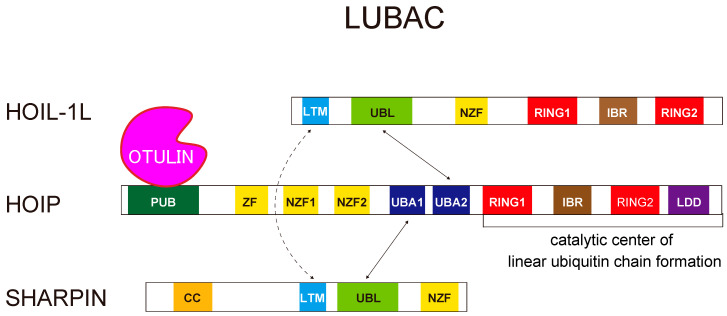
Schematic representation of the LUBAC ubiquitin ligase complex. LUBAC is composed of HOIL-1L, HOIP, and SHARPIN. HOIP interacts with the UBL domains of the other two components. The UBL domains of HOIL-1L interact with the UBA2 domain of HOIP, and SHARPIN UBL interacts with HOIP UBA1. Furthermore, both HOIL-1L and SHARPIN have LTM domains that fold into a single globular domain.

**Figure 4 cells-10-02706-f004:**
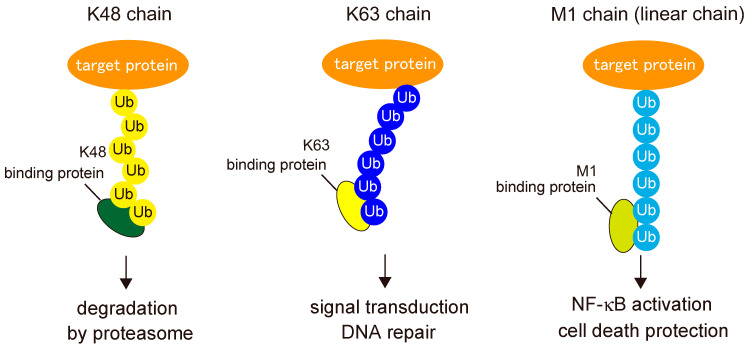
“Readers” of ubiquitin signals. Ubiquitin chains are decoded by specific binding proteins, “readers”, to mediate the specific functions according to each type of ubiquitin linkage.

**Figure 5 cells-10-02706-f005:**
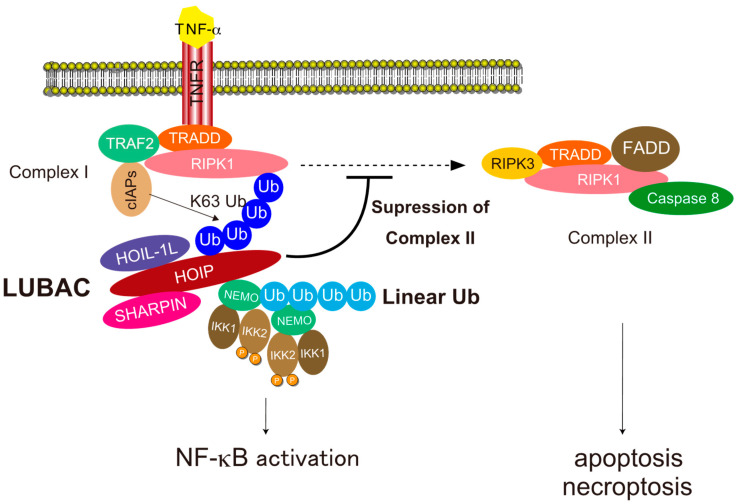
The mechanism underlying NF-κB activation and cell death suppression by LUBAC-mediated linear ubiquitination in TNF-α signaling. LUBAC is recruited to the activated TNF receptor complex via recognition of ubiquitin chains generated by other E3s and conjugates linear chains to NEMO. Linear chains conjugated to NEMO are recognized by the UBAN domain of NEMO in another IKK complex, leading to activation of IKK2 by dimerization of the IKK complex, autophosphorylation of IKK2, and subsequent activation of NF-κB. Linear ubiquitination of components of the activated TNF receptor complex decreases the production of the death-inducible TNFR-complex-II containing RIPK1, RIPK3, FADD, TRADD, and caspase-8, thereby suppressing death receptor-induced apoptosis and necroptosis.

**Figure 6 cells-10-02706-f006:**
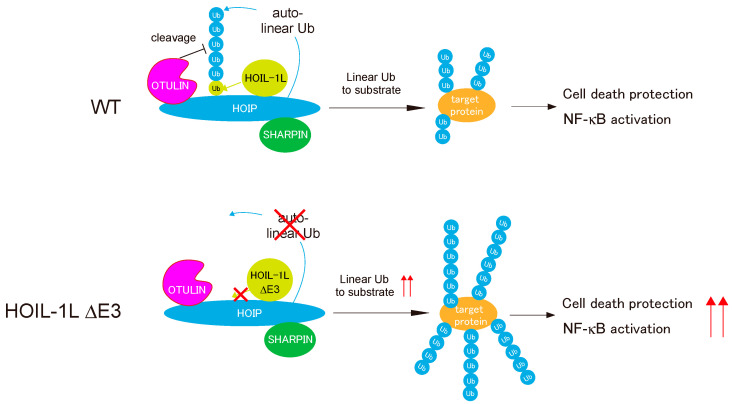
Regulation of the linear ubiquitination activity of LUBAC by HOIP, HOIL-1L, and OTULIN. HOIL-1L mono-ubiquitinates all LUBAC subunits (HOIL-1L, HOIP, and SHARPIN), and HOIP further conjugates linear ubiquitin chains to mono-ubiquitin, which is conjugated to LUBAC by HOIL-1L. OTULIN counteracts auto-linear ubiquitination of LUBAC. Loss of mono-ubiquitination of LUBAC following deletion of HOIL-1L E3 profoundly suppresses auto-linear ubiquitination of LUBAC and increases its linear ubiquitination activity towards substrates, activating the LUBAC functions of NF-κB activation and protecting against cell death.

**Figure 7 cells-10-02706-f007:**
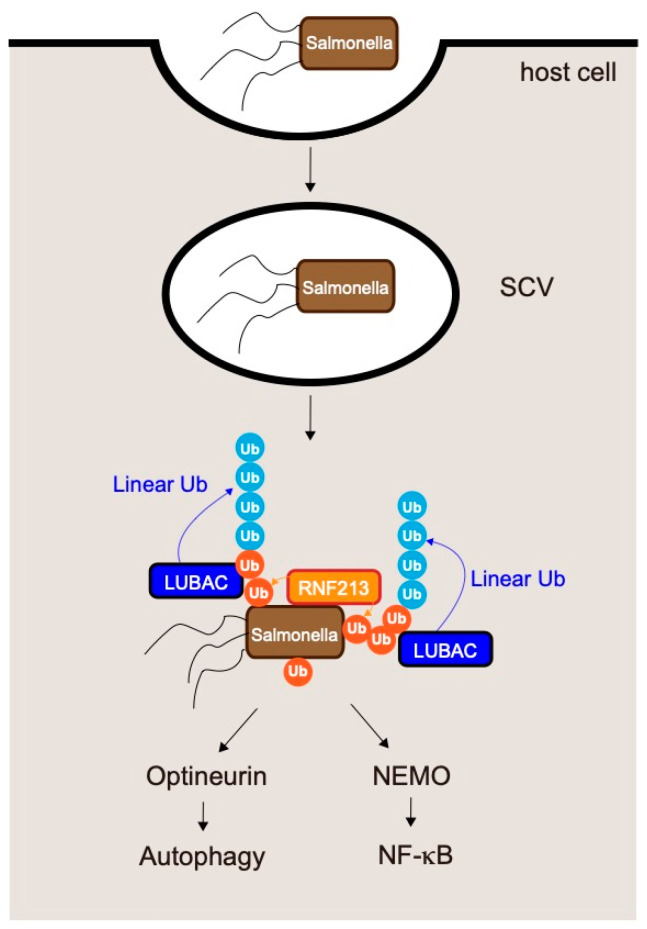
Mechanism underlying clearance of the intracellular pathogen *Salmonella* by LUBAC-mediated generation of linear ubiquitin chains. *Salmonella* invades epithelial cells; in the early phase of infection, the bacteria reside in *Salmonella*-containing vacuoles (SCVs). After several hours, the SCVs rupture and the *Salmonella* are exposed to the host cytosol. RNF213 directly conjugates ubiquitin to cytosolic *Salmonella*. Ubiquitination of *Salmonella* by RNF213 results in recruitment of LUBAC, which conjugates additional linear ubiquitin chains onto the ubiquitin added to the bacteria by RNF213. Linear chains conjugated by LUBAC restrict *Salmonella* proliferation by inducing xenophagy and NF-κB activation.

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
