# Peer review of "Biochemistry, Pathophysiology, and Regulation of Linear Ubiquitination: Intricate Regulation by Coordinated Functions of the Associated Ligase and Deubiquitinase"

_cells, 2021, doi:10.3390/cells10102706_

Round 1
Reviewer 1 Report
General comments: The review is very well written and contains a vast amount of information. However, for me it would need some more work especially in the cancer section which lacks, in term of clarity and information given, far behind the other parts.
Specific comments;
1) Abstract:
To my feeling the abstract goes to much right away into what LUBAC is. For readers not expert in ubiquitination it is complicated. Maybe a short recall what ubiquitination is and the existence of different types of chains might help to understand the Abstract better.
I also would prefer that things are made right way clear by saying that LUBAC is a ubiquitin ligase complex.
2) Figure 1: cell death protection seems to be written in a smaller typo.
3) Cancer part:
The 7.5. role of LUBAC in cancer
This part is not worked out enough, in part confusing and miss leading.
Authors say that Lubac activity is associated with carcinogenesis, which is a too general comment and authors should give some more details supporting this statement.
Authors say that it plays a role in resistance to cisplatin but nothing is described and published data show only that in the absence of HOIP cells are more sensitive to cisplatin. A critical review of the published data is needed.
And what about the axes Lubac - NFkB - cancer? In the introduction and later on authors point strongly to the role of LUBAC in NFkB signaling. It would therefore be very interresting and in part needed to discuss the Lubac - NFkB in cancer.
Similarly in the following part (8.1)
Authors state at the end of 8.1. that: " In addition to the crucial roles of LUBAC in the oncogenesis of ABC-DLBCL, LUBAC activity is also involved in resistance to immune checkpoint blockade therapy and cisplatin". Unfortunately, publish evidence supporting this statement is not given and needs to be discussed.
The same holds true for the following statement given by the authors: "The crucial role of LTM-mediated heterodimerization of the two accessory subunits in stable formation of trimeric LUBAC suggests a therapeutic strategy for the treatment of malignant tumors."
Yes but so what? Some explanation is missing.
So taken together the whole cancer part needs some more detailed work.
4) Section 8.2. Well, authors suggest that augmentation of LUBAC expression could be a way to fight infectious disease. However, as mentioned in 8.1 and 7.5: "augmentation of LUBAC is associated with carcinogenesis", how do the authors see this discrepancy?
5) Conclusion paragraph:
In the conclusion, authors mention again that deficiency of LUBAC components is associated with several disorders in humans. A table containing these different components and the associated disorders should be given.
Author Response
Point-by-point responses to the referees’ comments
Manuscript ID: cells-1382789
Reviewers’ comments
Point-by-point responses:
Reviewer #1
General comments: The review is very well written and contains a vast amount of information. However, for me it would need some more work especially in the cancer section which lacks, in term of clarity and information given, far behind the other parts.
Specific comments;
1) Abstract:
To my feeling the abstract goes to much right away into what LUBAC is. For readers not expert in ubiquitination it is complicated. Maybe a short recall what ubiquitination is and the existence of different types of chains might help to understand the Abstract better.
I also would prefer that things are made right way clear by saying that LUBAC is a ubiquitin ligase complex.
As suggested by the reviewer, we added the brief description of the ubiquitin system at the top of the abstract. Also, we described that LUBAC is a ubiquitin ligase complex in the abstract. (See lines 15–18 in the new manuscript).
2) Figure 1: cell death protection seems to be written in a smaller typo.
We appreciate the reviewer to point out our typo. We corrected the typo in Figure 1.
3) Cancer part:
The 7.5. role of LUBAC in cancer This part is not worked out enough, in part confusing and miss leading. Authors say that Lubac activity is associated with carcinogenesis, which is a too general comment and authors should give some more details supporting this statement. Authors say that it plays a role in resistance to cisplatin but nothing is described and published data show only that in the absence of HOIP cells are more sensitive to cisplatin. A critical review of the published data is needed.
And what about the axes Lubac - NFkB - cancer? In the introduction and later on authors point strongly to the role of LUBAC in NFkB signaling. It would therefore be very interesting and in part needed to discuss the Lubac - NFkB in cancer
We appreciate the reviewer for the constructive comment. As suggested by the reviewer, we modified the section 7.5 accordingly. We mentioned LUBAC-induced NF-kB activation in cancer as the authors suggested. In addition to NF-kB, protection from cell death is also involved in the carcinogenesis. We then referred to cell death protection.
Similarly in the following part (8.1) Authors state at the end of 8.1. that: " In addition to the crucial roles of LUBAC in the oncogenesis of ABC-DLBCL, LUBAC activity is also involved in resistance to immune checkpoint blockade therapy and cisplatin". Unfortunately, publish evidence supporting this statement is not given and needs to be discussed. The same holds true for the following statement given by the authors: "The crucial role of LTM-mediated heterodimerization of the two accessory subunits in stable formation of trimeric LUBAC suggests a therapeutic strategy for the treatment of malignant tumors." Yes but so what? Some explanation is missing. So taken together the whole cancer part needs some more detailed work.
As suggested by the reviewer, we modified the latter half of section 8.1. We described the involvement of LUBAC in immune checkpoint therapy in more detail. Also, we pointed out the importance of small molecules to inhibit LUBAC with minor side effect.
4) Section 8.2. Well, authors suggest that augmentation of LUBAC expression could be a way to fight infectious disease. However, as mentioned in 8.1 and 7.5: "augmentation of LUBAC is associated with carcinogenesis", how do the authors see this discrepancy?
We appreciate the reviewer for the thoughtful comment. We agree with the reviewer that augmentation of LUBAC to fight against pathogens might cause carcinogenesis. However, it seems unlikely because we found mice expressing a HOIL-1L mutant lacking E3 activity are viable up to the age of 12 months without overt phenotypes and augmented HOIP expression failed to induce lymphomagenesis, implying that agents that target the E3 activity of HOIL-1L could have fewer side effects. We address these points in the 8.2 section. (See lines 478-490 in the new manuscript).
5) Conclusion paragraph:
In the conclusion, authors mention again that deficiency of LUBAC components is associated with several disorders in humans. A table containing these different components and the associated disorders should be given.
As suggested by the reviewer, we added a table containing all LUBAC components and their associated disorders as Table 1.
Reviewer 2 Report
This is a comprehensive review about LUBAC regulation and function. The details are interestingly and correctly described. However, LUBAC regulation and function has been extensively reviewed over the last years, and to compete with the existing high-quality reviews this one need some polishing. Here are some comments and suggestions for improvement:
General comments:
- The title does not reflect the content of the review. Only a part of it describe ligases and deubiquitinases coordinate function, and this angle is not emphasised in the text.
- The first four sections are written in a different style than the last five which is comprised of short sections on different topics. They could be made more similar and the transition between the sections could better.
- Section 4 would read better before section 3. In this way all physiological functions would be in one place.
- As the review should describe the literature in an unbiased way, expressions like “We have identified” seems unappropriate
- Many facts are repeated very many times. Examples: “Among the subunits of the LUBAC ligase, HOIP is the catalytic center of linear ubiquitin chain formation” and “LUBAC is composed of three subunits: HOIL-1L, HOIP and SHARPIN.”
- Some parts lack references.
- All abbreviations need to be opened. Examples: SHANK, RING, UBAN, NZF
Comments on the sections:
Section 1.
- The introduction is not an introduction to the review, but an overview of ubiquitination in general, followed by a repeat of the exact wording of the abstract. Either sections should be rewritten.
- “We have identified a brand new ubiquitin chain” is not appropriate for a 15 year old discovery.
Section 2.2.
- It would be nice with more information of how the UBANs are more specific for M1-chains compared to other chain types
Section 2.3
- The first paragraph is confusing and unclear and should be rewritten.
- I miss a description on how LUBAC constitutively interacts with OTULIN and CYLD
- Add references to the last part of this section
Section 3.1
- Here many things are stated without references, especially in the last part. Many of the references are placed wrong and should be corrected.
Section 3.2
- The use of the word convert in the second sentence is confusing and could be changed
Section 5
- A recent study is in singular, but authors are referring to 2 studies
- The sentence “Linear ubiquitin chains can be conjugated to all LUBAC subunits, inhibiting the linear ubiquitination activity of the complex” is misleading, and would be clearified with an addition of “on other substrates”.
Section 6.2
- Section 6.2. is very short and a bit out of context. The information could be included in one of the other sections.
Section 7.2
- The phenotype of HOIL-1L-deficience is indicated to be mentioned above, but this is not found previously in the text.
Section 9
- This sentence in the middle of the section refering to the authors own work (without a reference) could be remover or rephrased: “We wondered ‘Why LUBAC had two different ubiquitin ligases in one E3 ligase complex?’ and unexpectedly found that the E3 activity of HOIL-1L plays a crucial role in LUBAC regulation.”
Comments on Figures:
- Throughout all figures, same colour code could be used for same chain types, for instance yellow for M1 and blue for K63
- The font sizes are different in Figure 1
- Typo in Figure 2: modifired
- Figure 5: label K63 chains, similarly as the linear chains
- Figure legend 5 and 6: TNFalpha doesn’t show properly and kappa is a “k” in NF-kappaB
- Figure 7 insinuates that LUBAC is bound to the bacteria and conjugate chains on mono-Ub, and not that LUBAC is recruited by ubiquitin chains as stated in the text (“LUBAC is recruited to Salmonella by recognizing ubiquitin chains on the bacteria and then conjugates linear ubiquitin chains to the preexisting ubiquitins.”)
Author Response
Point-by-point responses to the referees’ comments
Manuscript ID: cells-1382789
Reviewers’ comments
Point-by-point responses:
Reviewer #2
This is a comprehensive review about LUBAC regulation and function. The details are interestingly and correctly described. However, LUBAC regulation and function has been extensively reviewed over the last years, and to compete with the existing high-quality reviews this one need some polishing. Here are some comments and suggestions for improvement:
General comments:
- The title does not reflect the content of the review. Only a part of it describe ligases and deubiquitinases coordinate function, and this angle is not emphasised in the text.
As suggested by the reviewer, we modified the title of the manuscript. It is “Biochemistry, pathophysiology, and regulation of linear ubiquitination: intricate regulation by coordinated functions of the associated ligase and deubiquitinase”.
- The first four sections are written in a different style than the last five which is comprised of short sections on different topics. They could be made more similar and the transition between the sections could better.
As suggested by the reviewer, we combined section 6.2 with 6.3 to make the section more similar to the other sections.
- Section 4 would read better before section 3. In this way all physiological functions would be in one place.
We appreciate the reviewer for the constructive comment. We modified the two sections as the reviewer suggested.
- As the review should describe the literature in an unbiased way, expressions like “We have identified” seems unappropriate
We appreciate the reviewer for this important comment. As suggested, we removed phrases like “We have identified” from this manuscript.
- Many facts are repeated very many times. Examples: “Among the subunits of the LUBAC ligase, HOIP is the catalytic center of linear ubiquitin chain formation” and “LUBAC is composed of three subunits: HOIL-1L, HOIP and SHARPIN.”
As suggested by the reviewer, we removed these kinds of repeats as many as possible.
- Some parts lack references.
As suggested by the reviewer, we added references in some parts.
- All abbreviations need to be opened. Examples: SHANK, RING, UBAN, NZF
As suggested by the reviewer, we added SHANK, RING, UBAN, NZF in the abbreviation section.
Comments on the sections:
Section 1.
- The introduction is not an introduction to the review, but an overview of ubiquitination in general, followed by a repeat of the exact wording of the abstract. Either sections should be rewritten.
We appreciate the reviewer for the insightful comment. We modified the instruction part to introduce the ubiquitin system and reduced the description of linear chains to avoid the repeat of the abstract.
- “We have identified a brand new ubiquitin chain” is not appropriate for a 15 year old discovery.
As suggested by the reviewer, we removed “brand new” in that sentence and modified the sentence accordingly. (See lines 58 in the new manuscript).
Section 2.2.
- It would be nice with more information of how the UBANs are more specific for M1-chains compared to other chain types
We appreciate the reviewer for the thoughtful comment. NEMO UBAN recognizes the hydrophobic patches centered at Ile44 and Phe4 of the distal and proximal parts of linear ubiquitin respectively. We inserted a sentence in line 112-113 in the new manuscript.
Section 2.3
- The first paragraph is confusing and unclear and should be rewritten.
As suggested by the reviewer, we modified the first paragraph of section 2.3. (See lines 142-150 in the new manuscript).
- I miss a description on how LUBAC constitutively interacts with OTULIN and CYLD
We appreciate the reviewer for this important comment. As pointed out by the reviewer, interaction of CYLD or OTULIN with LUBAC is mutually exclusive. We then modified sentences accordingly. (See lines 151-158 in the new manuscript).
- Add references to the last part of this section
As suggested by the reviewer, we added references to the last part of this section. (See lines 172 in the new manuscript).
Section 3.1
- Here many things are stated without references, especially in the last part. Many of the references are placed wrong and should be corrected.
As suggested by the reviewer, we added and corrected references in this section (new section 4.1).
Section 3.2
- The use of the word convert in the second sentence is confusing and could be changed
We appreciate the reviewer for the insightful comment. We modified the sentence to remove “convert”. (See lines 232-234 in the new manuscript).
Section 5
- A recent study is in singular, but authors are referring to 2 studies
We are sorry for this silly mistake. We changed a recent study to recent studies. (See lines 249 in the new manuscript).
- The sentence “Linear ubiquitin chains can be conjugated to all LUBAC subunits, inhibiting the linear ubiquitination activity of the complex” is misleading, and would be clarified with an addition of “on other substrates”.
We very much appreciate the reviewer for the constructive comment. As suggested by the reviewer, we modified the sentence to include “on other substrates”. Also, linear ubiquitination to other substrates is regarded as “trans-linear ubiquitination” and linear ubiquitination to LUBAC is regarded as “auto-linear ubiquitination”. We inserted the phrases in a couple of part in section 5.
Section 6.2
- Section 6.2. is very short and a bit out of context. The information could be included in one of the other sections.
As suggested by the reviewer, we combined old Section 6.2 and Section 6.3 into new Section 6.2. (See lines 337-364 in the new manuscript).
Section 7.2
- The phenotype of HOIL-1L-deficience is indicated to be mentioned above, but this is not found previously in the text.
We are sorry for our mistake. We removed “As mentioned above,” in the second sentence in section 7.2. (See lines 383 in the new manuscript).
Section 9
- This sentence in the middle of the section referring to the authors own work (without a reference) could be remover or rephrased: “We wondered ‘Why LUBAC had two different ubiquitin ligases in one E3 ligase complex?’ and unexpectedly found that the E3 activity of HOIL-1L plays a crucial role in LUBAC regulation.”
As suggested by the reviewer, we removed the sentence. (See line 489 in the new manuscript).
Comments on Figures:
Throughout all figures, same colour code could be used for same chain types, for instance yellow for M1 and blue for K63
As suggested by the reviewer, we used the same color for the same chain types throughout all figures.
The font sizes are different in Figure 1
As suggested by the reviewer, we adjusted the font sizes in Figure1.
Typo in Figure 2: modifired
As suggested by the reviewer, we corrected the typo.
Figure 5: label K63 chains, similarly as the linear chains
As suggested by the reviewer, we also labeled K63.
Figure legend 5 and 6: TNFalpha doesn’t show properly and kappa is a “k” in NF-kappaB
As pointed out by the reviewer, we modified the italic fonts in the legends of Figures 5 and 6.
Figure 7 insinuates that LUBAC is bound to the bacteria and conjugate chains on mono-Ub, and not that LUBAC is recruited by ubiquitin chains as stated in the text (“LUBAC is recruited to Salmonella by recognizing ubiquitin chains on the bacteria and then conjugates linear ubiquitin chains to the preexisting ubiquitins.”)
We appreciate the reviewer for the important comment. Otten et al. indicated in ref 80 that in the presence of RNF213, ubiquitin chains are conjugated to the bacteria and LUBAC conjugated linear ubiquitin chains to salmonella. We then modified Figure 7 accordingly.
Round 2
Reviewer 1 Report
I thank the authors for considering the suggestions and for the corrections.